# Water IoT Monitoring System for Aquaponics Health and Fishery Applications

**DOI:** 10.3390/s22197679

**Published:** 2022-10-10

**Authors:** Mohammad Alselek, Jose M. Alcaraz-Calero, Jaume Segura-Garcia, Qi Wang

**Affiliations:** 1Paisley Campus, University of the West of Scotland, Paisley PA1 2BE, Scotland, UK; 2Computer Science Department, Escola Tecnica Superior d’Enginyeria, Universitat de Valencia, 46100 Burjassot, Spain

**Keywords:** aquaponics health, water IoT monitoring, WSN, LoRa, 5G

## Abstract

Aquaponic health is a very important in the food industry field, as currently there is a huge amount of fishing farms, and the demands are growing in the whole world. This work examines the process of developing an innovative aquaponics health monitoring system that incorporates high-tech back-end innovation sensors to examine fish and crop health and a data analytics framework with a low-tech front-end approach to feedback actions to farmers. The developed system improves the state-of-the-art in terms of aquaponics life cycle monitoring metrics and communication technologies, and the energy consumption has been reduced to make a sustainable system.

## 1. Introduction

Aquaponics and aquaculture has received growing concern in modern times, and this reinforces its increasing impact on society as an innovative response to food security [1]. Technological advancements in agriculture have opened a path for modern agricultural practices such as aquaponics, which is an eco-friendly system for cultivating fish and plants without soil [2]. Aquaponics is a symbiotic system which combines aquaculture with hydroponics in fish and plant cultivation to save water and utilize dissolved nutrients more efficiently, without the use of agrochemicals [3]. This advancement has also opened a new era for the design and development of both fish and plant health monitoring systems [2].

The time consumed in monitoring and resetting parameters to prerequisite levels is a major shortcoming in aquaponics systems [1]. Fish conserve flings, and it is tedious to replenish oxygen, maintain temperature, and water levels or fertilize plants frequently; hence, these metrics need to be frequently monitored to ensure fish and plant health or growth [3]. To overcome this complex and time-consuming process of manually monitoring plants and fish health in an aquaponic system, an automatic health monitoring sensor has been proposed. This study designs a smart health monitoring system which makes it easy for deployed sensors to accurately measure and transmit data on a variety of metrics such as pH, temperature, water level, and dissolved oxygen, etc., to remote farmhouse monitors in real-time.

The competitiveness of several developed aquaponics monitoring systems depends on technological advances, the number of metrics the system can monitor, the flexibility of design, cost-effectiveness, and differences in geographic environments, which cannot be generalised [1]. A lot of studies have been conducted on the design of aquaponics monitoring systems which can measure and display parameters such as pH level, water level, humidity, temperature, etc., to the user [4,5]. A few sensors, such as Libelium smart water ions, exist on the market. However, there is no cost-effective aquaponics monitoring system designed to detect selective ions in an industrially relevant aquaponics system. Existing designs are expensive and are mostly useful for laboratory use and not production or industrial use, and this has motivated the effort of this work.

### 1.1. Contributions

In this paper, the main contributions are related to the development of a complete 5G-enabled IoT system for fully monitoring the performance of fishery farms. Specifically, this IoT-based system improves the state-of-the-art and contributes to the improvement in the existing systems in the following aspects:Wide-Area-Enabled communications (>1 km^2^—using NB-IoT, LTE-M, and LoRa/ LoRaWAN);The system combines a higher number of sensors, compared to the state-of-the-art, in order to better characterize the studied environment;The system allows real-time monitoring and low-power consumption, as a sensor switching protocol (I2C-based) has been improved to perform more efficiently;The system is cost-efficient and allows for high modularity, as it has been implemented with a Docker-based system to install and update firmware (with OTA updating) in each node and in the farmhouse.

### 1.2. Paper Organization

The paper is developed as follows: The introductory section motivates the paper and the aim of the work. Section 2 (Materials and Methods) explains the context of our problem and the metrics needed. It also establishes the requirements and the architecture. Section 3 (Results) exposes the final design and some results in some fisheries, discussing the results according to the initial planning. The conclusion section summarizes the main achievements and the results.

## 2. Related Work

The state-of-the-art in automated aquaculture systems is somehow limited to the last years, when the capabilities of IoT systems and the increasing power of AI have allowed the creation of systems capable of monitoring, predicting states, and controlling the whole processes described in the cycle between the fishes and the plants.

In [6], the authors developed a system to monitor water quality, through Potential of Hydrogen (pH), sending the information with GSM in an Arduino-based architecture. Also in [7], the authors focused their efforts on the development of a low-cost, low-power, wireless sensor network and Arduino-based IoT system for groundwater monitoring. This system used LoRa for communications.

In [8], authors oriented their efforts to develop a system for water quality monitoring in remote locations (with total dissolved solids, electrical conductivity (EC), pH, and temperature sensors with LoRa-enabled communications and GPS). They also used LoRa for localization purposes and performed consumption measurements of their system.

In [9], the authors developed an interdigital FR4-based capacitive sensor, together with the whole IoT Arduino-based system (with LoRa and WiFi communications) to monitor nitrate concentration in water-monitoring studies.

In [10], the authors show an IoT system similar to the purpose of this project, applying the architecture for aquaculture purposes. This work shows a number of parameters lower than in our application, and their connectivity is based on WiFi (with ESP8266).

Additionally, the on-going H2020 project FutureEUAqua [11] is developing tools oriented to monitor, control, and manage the conventional aquaculture of major fish farming industries, promoting innovations in different productive areas. This project has a monitoring part, with wireless sensors in the fish farm, but the number of sensors is limited to dissolved oxygen (DO), salinity, chlorophyll, blue-green algae, and turbidity (based on real-time monitoring devices by InnovaSEA company [12]).

### Analysis of Existing Remote IoT Sensors Systems

According to one of our studies, the current state-of-the-art in the market shows a wide and diverse panorama of systems. At first glance, there is a clear market separation between self-contained devices which provide monitoring information on the screen of the device about the sensors attached to such devices and devices with wireless capabilities that are sending information to a central system. In the self-contained device category, there is a significant market clearly differentiated between those that support the measurement of concrete ions (e.g., $NH_{4}^{+}$, $NO_{2}^{-}$, $NO_{3}^{-}$) and those that provide more standardised values (pH, ORP, Temp, EC). Table 1 is a comparison table of the existing sensors under this category and their market price. Note that the DO feature pushed the prices up significantly.

In the wireless category, the state-of-the-art is significantly reduced, as shown in Table 2 below. From the state-of-the-art analysis, it can be concluded that there is no water monitoring solution from the analysed ones above specifically designed for aquaponics, and prices are completely prohibitive when wireless connectivity is required for centralised monitoring. In addition, the number of solutions for detecting selective ions are incredibly scarce (Libelium Smart Water Ions) if we consider those with wireless capabilities.

The implementation of this research work using cost-off-the-shelf solutions to demonstrate the concept would require a price per sensor unit of GBP 3800 for the minimally viable solution, provided by Libelium Smart Water, and GBP 9918 for the optimal solution provided by the combination of Libelium Smart Water and Smart Water Ions. The prices available in Table 1 and Table 2 have been collected in June 2022.

This high cost has motivated the efforts of this work. The efforts in terms of research, development, and innovation have been focused on providing a new, cost-effective, competitive solution:Specifically designed for aquaponics;Designed with flexibility to accommodate different market ranges and use cases (cheaper but less accurate versions and more expensive but more accurate versions);More cost-effective than any of the existing competitors.

Meanwhile, ready-to-buy sensors have been acquired to deliver the demonstration part of the project while efforts have been put into developing an alternative to those sensors. In terms of design and demonstration purposes for the health monitoring system, only one of the plant-growing beds and fish-growing areas has been instrumented in order to deal with complexity and costing. The final cost of our proposal for the complete node is GBP 1500 (considering all the sensor probes, the electronics, and the communications).

## 3. Materials and Methods

The aquaponics system comprises the dynamic interaction of fish, plants, bacteria, and their aqueous environment [1]. The main intention of an aquaponic system is to reuse nutrients being pulled out of fishes which are necessary for growing plants [1]. Fish are grown in a fishpond, and they are regularly fed. Along their vital cycle, they produce decompositions that are rich in ammonium ($NH_{4}^{+}$). Such decompositions will travel along the pipeline system into the growing bed with plants and bacteria in the soil, as shown in Figure 1 below, which depicts the basic principles of an aquaponic system. The bacteria in the soil transforms ammonium ions ($NH_{4}^{+}$) firstly into nitrites ($NO_{2}^{-}$) then nitrates ($NO_{3}^{-}$). Nitrates ($NO_{3}^{-}$) are absorbed by the plants as part of their vital cycle, then water is oxygenated ($H_{2}O$) and filtered off. To close the loop, the oxygenated water can be returned to the fishponds, and depending on the size of the fishpond and the growing beds, one or two additional filters may need to be inserted along the cycle. First, a biofilter could be in the segment between fish decomposition and plants. This filter is not habitable by bacteria that facilitate the transformation from ammonium ($NH_{4}^{+}$) to nitrates ($NO_{3}^{-}$); hence, it could be inserted to help control the population of bacteria in the system. Second, a Swirl filter could be placed in the segment between the plant evacuation and fishes’ intake of water. The purpose of this filter is to purify the water by eliminating solid materials, thereby making it suitable for fish life. This solid material is usually present when the plants cannot provide sufficient filtering (i.e., due to different parameters, e.g., size of the bed, number of fish, etc.).

### 3.1. Water Monitoring Parameters

There is a set of water probes which can be used to build a digital health monitoring system. Most probes are used for the detection of one or more water metrics such as pH, temperature, and ammonia, etc. These probes consist of two interfaces; the first interface reacts with water, based on physical or chemical principles to produce the values of the metric(s). The second interface is an electrical interface used to allow a digital system to retrieve the measured metrics. The key aspect to consider in the selection of water probes are:The time elapsed between each required calibration;The accuracy of the measured metric;The duration until water probe is not functional (lasting period).

These aspects directly influence the operational aspects of the deployed system. Other aspects such as the electrical interface have more engineering requirements, which will directly affect the final cost of the system. The number of metrics to be monitored will also have an impact on the quality and cost of the developed monitoring system, since each monitored metric/parameter will have an associated direct cost. There is a non-exhaustive list of water probes in the market which can be used to enhance the capabilities of the digital health monitoring system for aquaponics.

The following are parameters or metrics for monitoring water, soil, and air quality. For water quality monitoring:Potential of Hydrogen (pH): this is a scale used to specify the acidity or basicity of an aqueous solution [22];Water Temperature: temperature of the water;Dissolved Oxygen (DO): this is the amount of gaseous oxygen ($O_{2}$) dissolved in the water [23];Oxidation-Reduction Potential (ORP): this is a measure of the tendency of a chemical specie to acquire electrons from or lose electrons to an electrode thereby reduced or oxydised, respectively [24];Electrical Conductivity (EC): this is the measure of the amount of electrical current a material can carry or its ability to carry a current [25];Total Dissolved Solids (TDS): this is a measure of the dissolved combined content of all inorganic and organic substances present in a liquid in molecular, ionised, or micro-granular (colloidal sol) suspended form [26];Total suspended solids (TSS): this is the dry weight of suspended particles that are not dissolved in a sample of water and that can be trapped by a filter that is analysed using a filtration apparatus [27];Water Level (WL): this is the level of water along a given surface (e.g., fishpond);Water Flow (WF): this is the speed of the water across the pipeline circuit;Ion-Selective Ammonia ($NH_{4}^{+}$): this is the concentration of $NH_{4}^{+}$ in aqueous solutions;Ion-Selective Nitrite ($NO_{2}^{-}$ ): this is the concentration of $NO_{2}^{-}$ in aqueous solutions;Ion-Selective Nitrate ($NO_{3}^{-}$): this is the concentration of $NO_{3}^{-}$ in aqueous solutions.

Here, it is worth emphasising that the hydroponics metrics (i.e., ORP, pH, EC, Temp) and the aquaponics metrics (i.e., $NH_{4}^{+}$, $NO_{2}^{-}$, and $NO_{3}^{-}$) are the main metrics to control the aquaponic cycle.

We have also considered other metrics for soil and air quality monitoring:Light Intensity (LI): it measures the number of lux to determine suitability to carry out photosynthesis processes in plants;Air Temperature (AT): it is the temperature of the air;Air Humidity (AH): it is the humidity of the air;Wind Direction and Speed (WDS): it is the speed of winds;Soil Moisture (SM): it is the level of water in-between soil particles.

### 3.2. Analysis of Requirements for Aquaponics System

The analysis of requirements for the aquaponics system has followed a two-fold methodology. The identification of the minimal requirements an aquaponics monitoring system should possess when optimizing for a low-cost monitoring system available for all markets and the identification of requirements needed to create an internationally competitive monitoring system. This methodology allows us to maximise the exploitative aspects of the outcomes generated in the project and optimize for the concrete system being addressed in the demonstration phase of the project. In this report, the two-fold methodology will be referred to as an “ideal scenario” and a “minimal suitable scenario”, respectively.

Figure 2 shows a conceptual simplified version of a floorplan representing an aquaponics system. It is composed by fish ponds, grow beds for vegetables, bio-filters and swirl filters, pumps and pipes to interconnect the whole system, and the points to connect the sensors to send information to the farmhouse.

The ideal scenario will allow for per-pound metrics, per-grow bed metrics, and per-filter metrics for each sensor available in the system; however, this represents a significant increase in price for the final product. This kind of system is conceptually viable but economically not affordable. Thus, the next step in the trade-off for accuracy/price would be the use of four sensors, i.e., 2× Type-A sensors and 2× Type B sensors, explained later. The system will alert the farmer if there is an issue, the type of issue, and where it can be fixed; however, determining the best place to fix the issue might require an in situ investigation outside of the data monitoring health system. This is what we have coined as the “ideal scenario”. The “minimally suitable” deployment would be to make use of two sensors (2× Type A+B sensors).

The Type A sensor is located immediately after the fishponds; thus, it measures the water parameters related to fish health. These parameters include the pH, EC, TDS, DO, temperature, $NH_{4}^{+}$ and/or $NH_{3}$, and/or ORP, and these parameters should be measured in an ideal scenario. In a minimally suitable scenario, $NH_{4}^{+}$ and $NH_{3}$ could be substituted for ORP, carrying out a trade-off between accuracy and cost. This minimally suitable scenario will allow the farmer to know if there is a higher $NH_{3}$ or $NH_{4}^{+}$ concentration. It is important to note that $NH_{3}$ is toxic for fish and can cause death, but $NH_{4}^{+}$ is not as toxic and can only cause death at very high concentrations. In addition, EC could be replaced by a cheaper and less accurate TDS sensor. EC measures the electrical conductivity of water, whereas TDS measures the level of particles dissolved in water, and both are good estimators of the number of ions present in water. This sensor also ensures that water reaching the fishponds is clear and compatible with fish health. Type B sensors are focused on providing the level of plan nutrients generated by the bio-filter to estimate the quality of the nutrients for the plants. Thus, it should provide pH, EC, $NO_{3}$, and $NO_{2}$ levels in an ideal scenario. In a minimally suitable scenario, $NO_{3}$ and $NO_{2}$ could be measured out of the data monitoring system while setting up the biofilter until it reaches expected levels of bacteria; then, EC will be used to estimate the concentration of ions (nutrients) in water. This sensor is also focused on determining the quality of the water being provided by the grow beds.

Table 3 represents the different types of sensors and their associated monitoring parameters. The table also contains a combined sensor with all the parameters together to save manufacturing time. Table 3 shows the minimal viable set of metrics, and for the minimal viable set of metrics, there are other metrics indicated with *. The metrics indicated with * are metrics which should be measured using a cheap colorimetric test in providing $NH_{4}$, $NO_{2}$, and $NO_{3}$ measures during the establishment phase of the aquaponics system. After that establishment, they will need to be measured manually, only if the EC and ORP values are not in a normal range.

In terms of communication, the floor plan is extended within an area greater than 200 m^2^, but the system should be designed to cover up to 1 km^2^, as the farmers control room, i.e., where the central system is located and where the sensors will be located, cannot be assumed; thus, an average coverage of around 1 km^2^ should be supported by the system.

The technical specifications for the sensors included in the node are shown in Table 4.

## 4. Proposed IoT System Architecture for Aquaponics Monitoring

The proposal and development of an IoT system architecture for monitoring aquaponics health is very useful for the fishery farms industry as the automation in the care and exploitation of fish allows for an improvement in production.

Figure 3 shows the proposed IoT system architecture for aquaponics monitoring. This system is comprised of a farmhouse which houses a monitor, WiFi link (or wireless technology), keyboard, mouse, and gateway box. The system also considers a growing bed; a 1 km^2^ wireless link; a sensor box with a set of sensors deployed in the field according to the analysis in Section 3.1, where different types of field sensors will be deployed in fishponds; a sump; and filters to monitor appropriate metrics. The farmhouse (as explained in Section 6) contains a gateway sensor, which forwards all the metrics from LoRa technology to other technologies and a monitoring station, which allows the first visualisation and control of the metrics. The gateway sensor also allows the connection to other infrastructures through WiFi or LTE-M/NB-IoT. This infrastructure allows the connection to the CORE and Cloud and the monitoring from a remote/house station.

LoRaWAN is the specification for Low-Power Wide-Area Networks (LPWAN) in Low Radiation (LoRa) communications. From the LoRa protocol, LoRaWAN is a point-to-multipoint network layer protocol, full duplex, and with the best LoRa features, i.e. low-power, extending battery duration and quality of service (with three modes), and uses FSK modulation. LoRa-based technology is able to achieve long ranges.

LTE-M (or LTE-MTC (Machine-Type Communication)), including eMTC (enhanced Machine-Type Communication), is a low-power wide-area network (LPWAN) radio technology standard developed by 3GPP to enable a wide range of cellular devices and services (specifically for machine-to-machine and Internet of Things applications). The specification for eMTC (LTE Cat M1) was frozen in 3GPP Release 13 (LTE Advanced Pro) in June 2016.

NB-IoT is also an LPWAN protocol, and it is a radio technology standard developed by 3GPP to enable high-end services for mobile devices. Previously, the 3GPP Release 13 (LTE Advanced Pro) specification was used until June 2016. NB-IoT uses an extension of the standard LTE network but limits bandwidth to a single 200 kHz band. It uses OFDM modulation for communications with few connections between devices and uses SC-FDMA for communications with more connections between devices.

The advantage of LTE-M over NB-IoT is its comparatively higher data rate, mobility, and voice over the network, but it requires more bandwidth and is more costly. Compared to an LTE Release 12 Cat-0 modem, an LTE-M modem is claimed to be 80% less expensive, supports up to 18 dB better coverage, and has a battery lifetime that can last up to several years.

The data link from the node to the farmhouse is made using the Message Queuing Telemetry Transport (MQTT) protocol. This protocol can be configured with 3 levels (i.e., 0, 1, and 2), but for our purpose, it has been configured with level 2 for both subscribers and publishers to ensure information delivery and no data duplication. The security issues were tackled by configuring the Transport Layer Security (TLS) encryption to ensure secure communication between clients and brokers.

For the sensor to monitor metrics, different water probes will be attached to the sensors. The design of the sensors should be flexible enough to allow the building of a system which can be customised with sensor combinations described in Table 3 as type A, type B, type A + B, or a minimal type A + B in order to control production costs. Each of these sensors has attached a wireless link which is able to provide coverage of up to 1 km^2^ in order to ensure that the solution is flexible enough to accommodate farmland extensions dedicated to aquaponics, such that the farmer can have a view of all his/her farmlands on the same monitoring system. LoRa/LoRaWAN, NB-IoT (5G), and LTE-M technologies will provide the capabilities to support this coverage. In the farmhouse, a special sensor called the gateway sensor will be deployed. The main purpose of this sensor is to act as a receiver for all the metrics periodically sent by all other devices, and it is also responsible for sending the calibration information to the associated sensor deployed in the field. The gateway sensor is connected to two different wireless networks, and it acts as a forwarder between them. Any information received by the LoRa/LoRaWAN network will be forwarded into the WiFi network and vice versa. Finally, a small computer with a monitor, keyboard, mouse, and WiFi connection will be installed in the farmhouse to allow the farmers see the graphical interface where all the information is displayed in real-time.

In our proposed architecture, we have designed and prototyped both the device and gateway capabilities indicated in the IoT Methodology standardised by ITU-T under specification ITU-T Y.2060 [28]. However, at this stage, we have not specified the definition of the functional components defined in ITU-T Y.4115-2017 [29], as our IoT system is an industrially relevant, lab-based development, so the IoT Device Capability Exposure (IoT DCE) is summarised in a Database and a Dashboard (in the Farmhouse subsystem) but could be replaced by a complete IoT ecosystem, such as Thingsboard, FIWARE, or any other platform.

## 5. In-pond WaterMon Sensor Design and Feasibility

Sensors are devices deployed in the aquaponics system to collect real-time data on temperature, humidity, pH, dissolved oxygen, total dissolved solids, water level, etc. A detailed design of the sensors referred to as WaterMon has been carried out to achieve a step closer to the market.


*Overview Design*


The design of the monitoring system has been thought to be implemented on a four-layer PCB board especially designed for the purposes of this work.

In Figure 4, the Fipy I2C connection is serially connected to the different sensors and the analog to signal converters. I2C, or the Inter-Integrated-Circuit protocol, is a synchronous, multi-controller/multi-target, packet-switched, single-ended, serial communication bus. It is widely used for attaching lower-speed peripheral ICs to processors and microcontrollers in short-distance, intra-board communication.

These devices have been configured by software to be switched on, read a measure from a sensor, and switch it off to go to the next device to collect its measurement. The sensor is selected by using a 74HC237 decoder multiplexer [30] and programmed in order to follow a specific order for the data collection within the sensors. The connection of the keypad to the I2C is oriented to write in the system the adequate calibration, which will be further explained in Section 5.1. In addition, the LCD connection will allow the visualisation of the information for every step of the calibration or the monitoring process.

WaterMon allows the monitoring of 11 different water quality metrics, including all of those indicated in Table 3. WaterMon has been designed to provide connectivity to LoRa, LoRaWAN, and the new 5G networks by supporting the novel IoT technologies, LTE-M and NB-IoT, and low-profile technologies such as WiFi and Bluetooth. For the implementation of the LoRa protocol, we have selected the parameters shown in Table 5.

The previous design has been originally prototyped in a lab-based implementation to test the feasibility of the sensor. Figure 5 shows an image captured with the early prototype achieved in the context of the project where five different water monitoring probes have been implemented as an early-stage feasibility (DO, EC, temperature, ORP, and pH).

From this early prototype, a fully functional optimised version of the prototype in a printed circuit board (PCB) was designed to achieve a better performance and next to a final product. Figure 6 shows a screenshot of some of the layers available in the prototyped sensor. This board has been squeezed in 20 × 14 cm to minimize fabrication costs.

Furthermore, after testing the methodology using the virtualised 3D version of the PCB, the manufacturing was carried out by an external company, and the result is a fully functional sensor board.

Afterwards, all the sensor components were soldered into the PCB, creating a fully functional prototype. The water monitoring probes were not attached before being physically inserted in liquid (as shown in the pictures) so as to not cause any damage, and thus, we put this on hold to perform the deployment of the probes in an industrially relevant scenario. The device allows the installation of 1 to 11 sensor probes, thus allowing for the customization of four different types of field sensors considered in Table 3 of this document (Type A, B, A + B, minimal A + B).

With respect to the software governing the behaviour of the WaterMon board, there is no open-source initiative which could be used to have a starting point. Therefore, an implementation of the software has been carried out until a fully functional prototype is achieved. This prototype allows us to:1.Perform the continuous monitoring of the 11 metrics supported in the board;2.Perform the continuous reporting of the values retrieved from the probe using the NB-IoT, LTE-M, LoRa/LoRaWAN wireless connection;3.Perform the calibration of the water monitoring probes in a very intuitive way.

The PCB has been designed with flexibility and extensibility in mind. Therefore, it allows for the attachment of more sensors, and this would propel further research in this field. The PCB also allows for the extension of capabilities such as controlling for the sensor’s energy consumption rate to enable future developments of battery-powered or solar-powered versions; however, software has not been developed to support such capabilities since this is out of the scope of this project. Finally, an enclosure has been designed to achieve a device ready to be deployed in an industrially relevant scenario.

Figure 7 and Figure 8 show the whole sensor device, fully operative to be used in a fish farm, with proper communications.

### 5.1. Calibration Procedure

The calibration process is performed using a one- or two-point calibration for each sensor probe. Figure 9 shows an illustration of the two-point calibration procedure. Following this calibration process for each sensor, the first step is to put the probe into the corresponding calibrating liquid and press the corresponding button on the keypad, which has been programmed to make the probe selection (for DO, EC, ORP, pH, and Temp) and write the calibration into the corresponding device. In the case of the sensors connected to 16 bits A/D converters ADS1115 (for $NH_{4}^{+}$, $NH_{3}$, $NO_{2}$, and TDS), the calibration is stored on the NVRAM in the Fipy.

#### Compensation

Once every sensing probe is calibrated, the node has been programmed to consider measurement compensation for each parameter. Table 6 shows the parameter dependencies for some sensors (the other ones not appearing in the table does not need any compensation). This process needs a specific measurement collection order, which has been considered by programming the previously mentioned 74HC237 decoder multiplexer to make the sensor selection.

When compensation is required by a particular sensor, it has the purpose to provide the highest possible accuracy in the measurement of the values measured by such sensor. Calibration, previously described, is the first approach towards the achievement of such accuracy by dealing with the aging of the sensor, the state of the chemical membranes and liquids used to carry out the physical measurement, and other aspect about the status of the sensor. After that, the compensation will allow one to tweak the values read from an already calibrated sensor in order to adapt it with the external factors that may affect the reading of such values. Thus, for example, EC in Table 6 is dependent on water temperature. It means that the value read about the electrical conductivity in water is directly affected by the temperature of the water, and thus, by knowing that value in advance of the reading of EC values, it can be used to compensate the read value in order to provide a much more accurate result.

## 6. Farmhouse System

### 6.1. Gateway Sensor

In order to extend the communication range as proposed in Figure 3, a gateway sensor has been proposed.

For the Gateway sensor, we will use a state-of-the-art, ready-to-use sensor, specifically Pycom FiPy [31] with PySense 2.0 by Pycom [32] extension boards. However, even if the hardware is currently mature enough, the software inside the sensor needed to perform the ’forwarding’ capabilities between LoRa/LoRaWAN and WiFi does not exist. This software has been fully developed in the context of this work to produce a fully functionally sensor. Figure 10 shows an image of the prototyped gateway. The prototype has two different antennas to connect two different wireless technologies, and it has a USB cable mainly used to provide power to the device (and program the software when required). Figure 11 also shows the PyCase enclosure bought for the device to create a ready-to-deploy product.

### 6.2. Monitoring Station

In terms of hardware, we are relying on a very cheap (70 GBP) commercially available mini-computer, Raspberry Pi 4B+ (RPi4), used together with an SD card, a keyboard, a mouse, and a monitor to provide a fully functional computer with basic capabilities, which is more than enough to fully fulfill the requirement of the use case. Figure 12 shows the schema of the system located in the farmhouse and the connections for each subsystem oriented to the hardware deployment. This image is the setup achieved on our premises. This hardware will be situated in the farm’s control and management centre, which consists of the mini-computer, Raspberry Pi 4, with an SD card, a keyboard, a monitor, and a mouse to monitor and control the aquaponics system remotely.

In terms of the software to show the real-time data monitoring information of the whole system, several monitoring tools are available on the market. We have been customizing Grafana as the tool to display the sensor metrics on screen. Figure 13 shows the graphical interface shown to farmers on the mini-computer. Using displayed or obtained information, the farmer decides whether to open the air pump, water pump, lights, and feeder or make other changes required to sustain fish and plant health.

## 7. Results and Discussion

The development of this IoT-based aquaponic system has a number of future applications in the fishery industry. One of the first tests conducted is the functional testing, which has allowed us to achieve good performance in a local and a networked environment. This testing has been successful and has shown good performance for the communication and the sensing parts.

As a proof of functioning and performance, the first test of the WaterMon node is the time measurement for every sensor sending information to the farmhouse. Figure 14 shows that the most time-consuming measurement is DO, which takes an average time of 1.5 s for every measurement and sending cycle; the pH and ORP sensors take an average time of 1.1 s approximately; and the EC and RTD sensors take 0.9 s approximately. This fact is mostly due to the need of some compensations for the measurements. The analog sensors are more fluent in their measurements, with 0.3 s each.

In order to evaluate the whole monitoring cycle in the WaterMon node, a test with the node just collecting cycles of parameters has been conducted. Figure 15 shows a plot of the time used for each cycle, which takes, on average, 7430 ms (with a standard deviation of 1.6 ms).

Figure 16 shows the timing used to submit the information to the farmhouse from the values collected at the node. For a number of monitoring cycle submissions, the average value is 462.7 ms (with a standard deviation of 56.9 ms).

Finally, an energetic consumption test has been conducted to assess the effectiveness of the energy-aware implementation for this system. Figure 17 shows the power consumption measurement during 100 s of the WaterMon sensor. As it can be seen, the average power reduction for the situation with and without the IC 74HC237 is 0.76 W in 100 s, when we use the IC. This reduction allows us to extend the lifetime of the power supply for this purpose. For instance, if the power supply is batteries, a commercial power-bank with 20,000 mAh will give the WaterMon sensor without the IC a lifetime of 41.2 h, while if we consider the use of the IC, the lifetime is extended up to 59.8 h, an increase of the power-supply time of 68.9%.

## 8. Conclusions

This paper summarises the work performed in the development and testing of a whole system for aquaponics health monitoring oriented to the fishery industry. We have checked that the existing monitoring systems in the state-of-the-art do not fit the requirements for monitoring the complete aquaponics life cycle.

The proposed system improves the state-of-the-art in terms of communication technology as it involves 5G communications, but for long-range communication, it uses LoRa in the nodes and Gateways with LTE-M/NB-IoT to establish Internet connectivity.

For aquaculture purposes, the sensing layer has been improved to include a number of sensors to improve the performance in the fishery industry. The most time-consuming processes are linked to the sensing and compensation processes, which range between 0.9 s to 1.5 s for the digital sensors and 0.3 s for the analog ones. The whole sensing period takes around 7.5 s on average, with nine sensors. The mean time for each information packet to reach the CORE from the sensing node is around 463 ms.

The hardware architecture has also been designed to improve the energetic performance and reduce power consumption, allowing an extension of the battery life of around 70%, with the inclusion of a switching IC for the sensors (74HC237) connected to the I2C line.

## Figures and Tables

**Figure 1 sensors-22-07679-f001:**
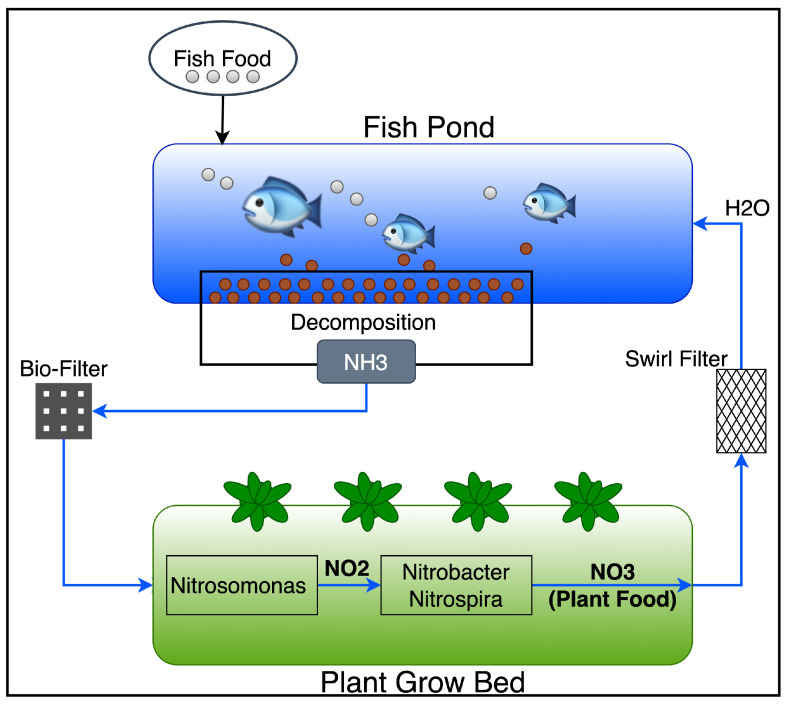
Aquaponics cycle.

**Figure 2 sensors-22-07679-f002:**
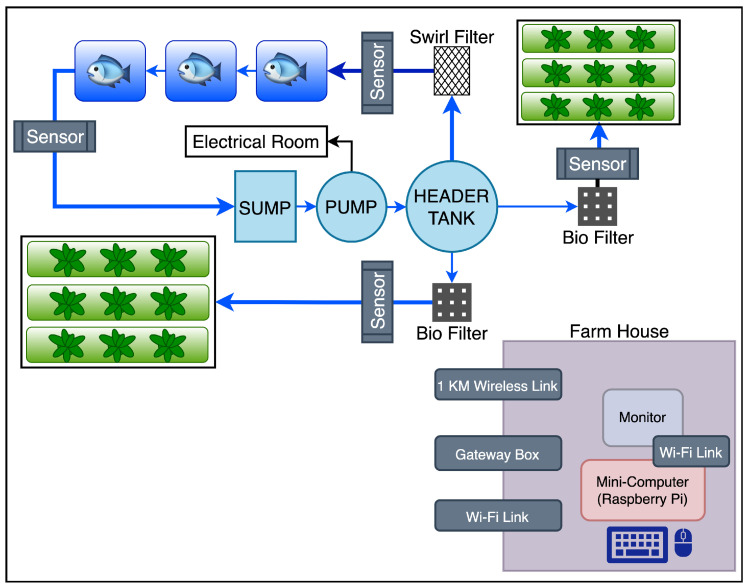
Simple aquaponics system floor plan.

**Figure 3 sensors-22-07679-f003:**
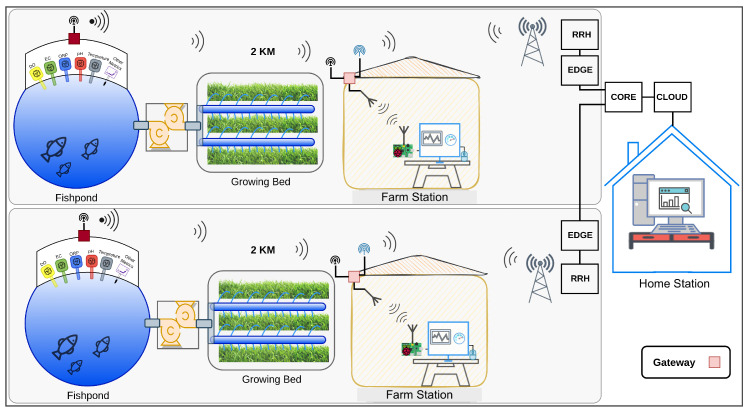
Proposed architecture for our Aquaponics Monitoring System.

**Figure 4 sensors-22-07679-f004:**
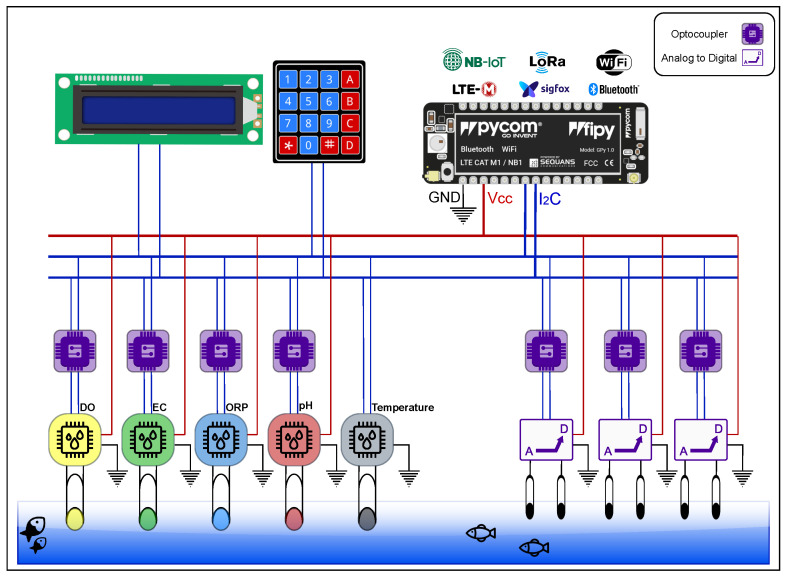
Overview Design of WaterMon System.

**Figure 5 sensors-22-07679-f005:**
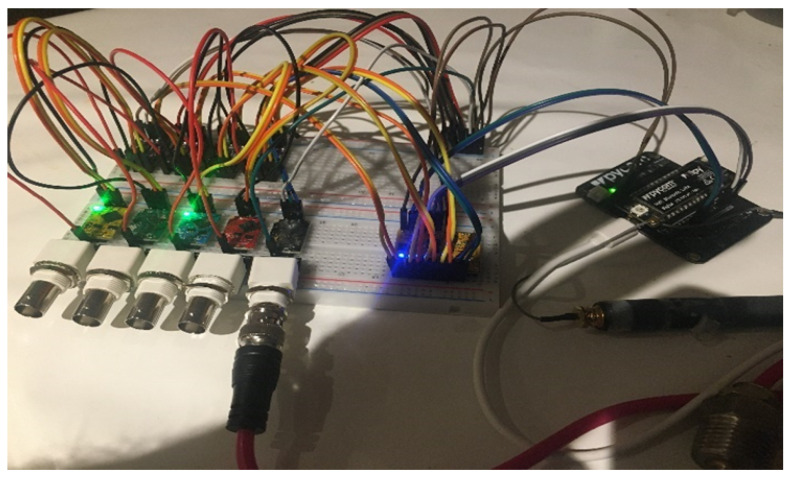
Early Prototype.

**Figure 6 sensors-22-07679-f006:**
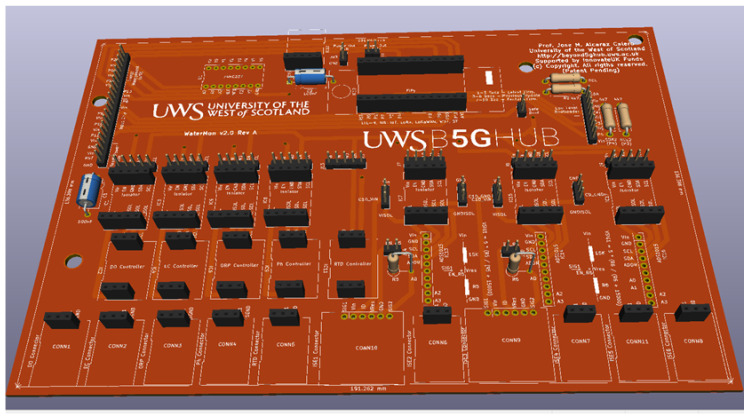
Three-dimensional Representation of PCB.

**Figure 7 sensors-22-07679-f007:**
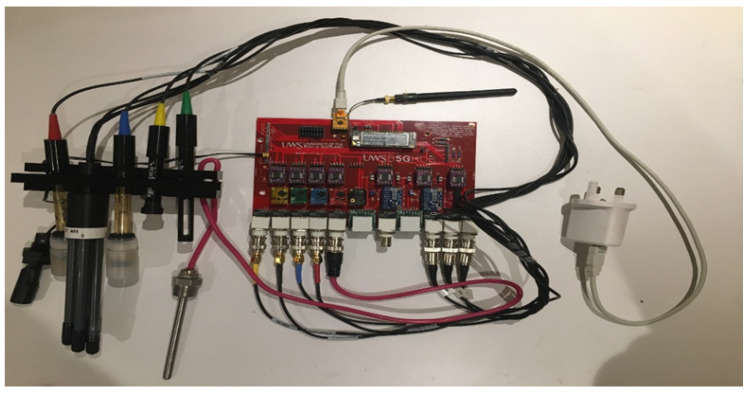
WaterMon Sensor.

**Figure 8 sensors-22-07679-f008:**
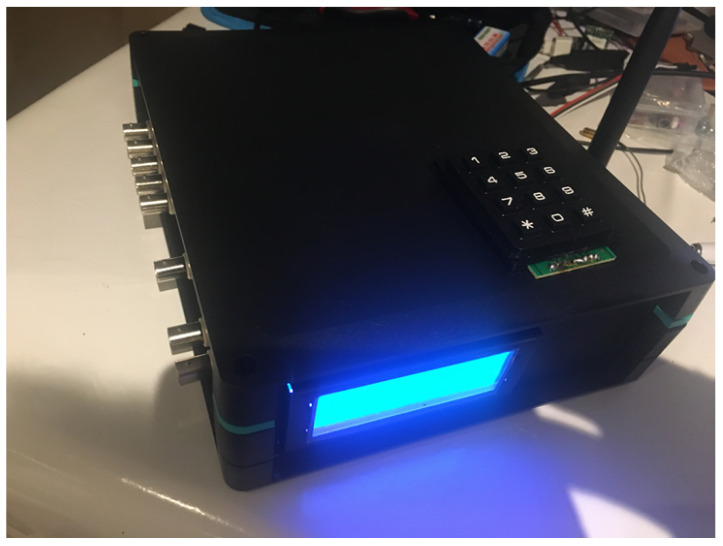
Sensor Device.

**Figure 9 sensors-22-07679-f009:**
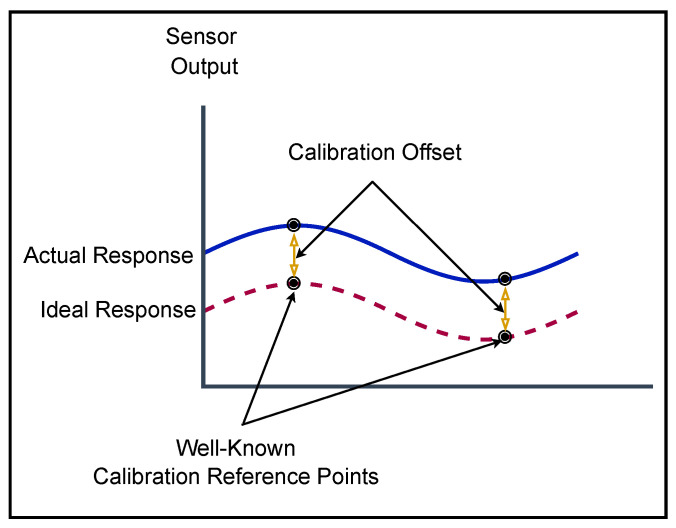
Sensor Calibration.

**Figure 10 sensors-22-07679-f010:**
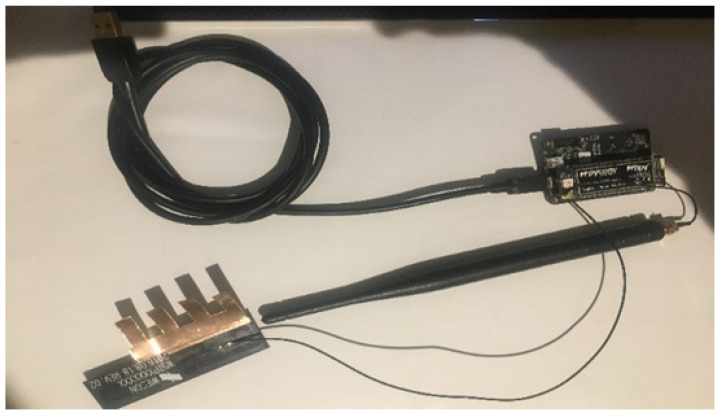
Prototyped Gateway.

**Figure 11 sensors-22-07679-f011:**
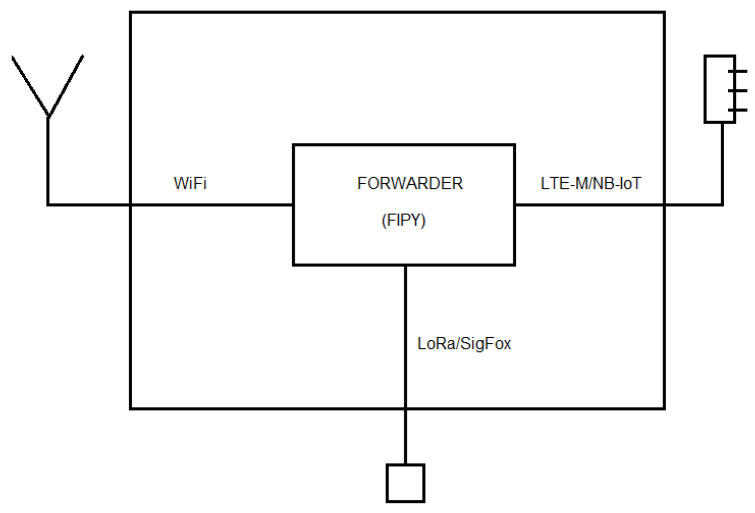
Gateway diagram.

**Figure 12 sensors-22-07679-f012:**
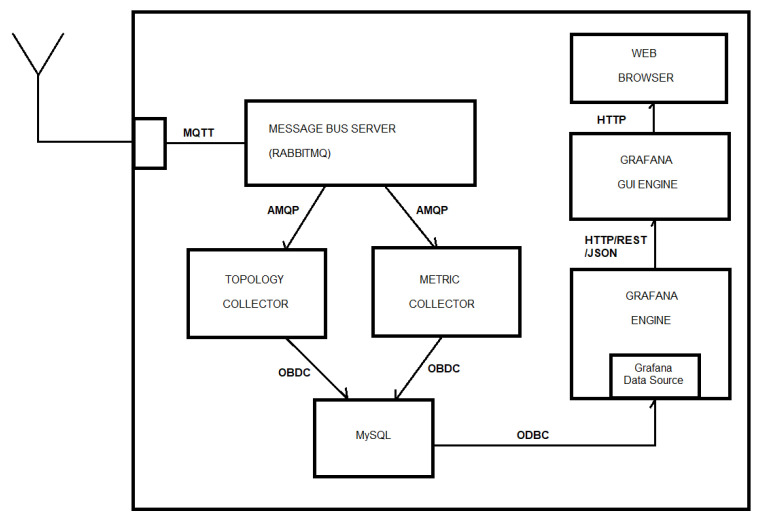
Farmhouse system schema.

**Figure 13 sensors-22-07679-f013:**
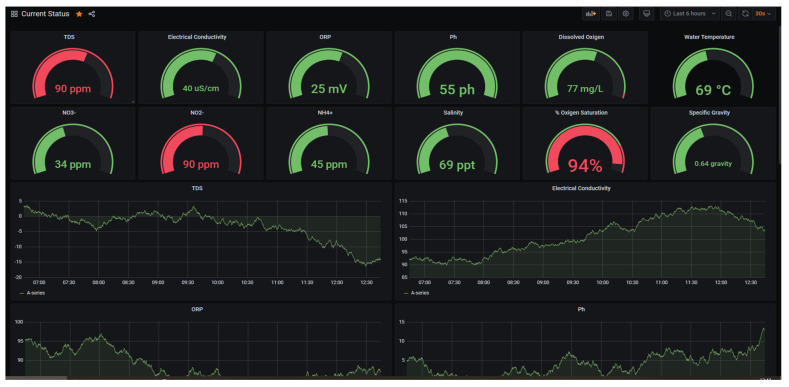
Graphical interface on RPi4.

**Figure 14 sensors-22-07679-f014:**
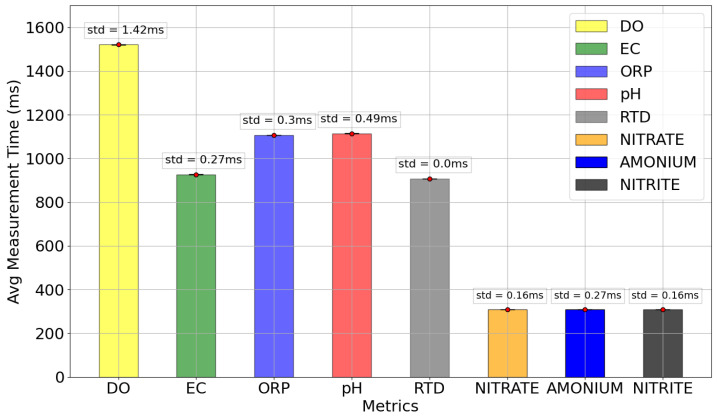
Time Measurements of WaterMon Sensors.

**Figure 15 sensors-22-07679-f015:**
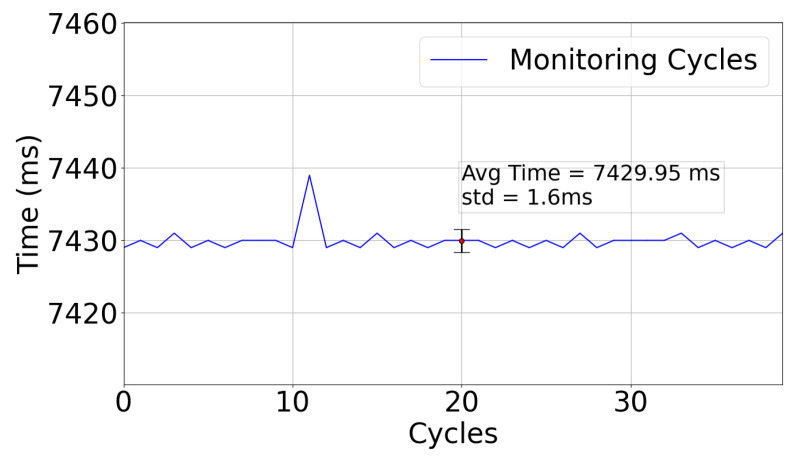
Monitoring Cycle Time.

**Figure 16 sensors-22-07679-f016:**
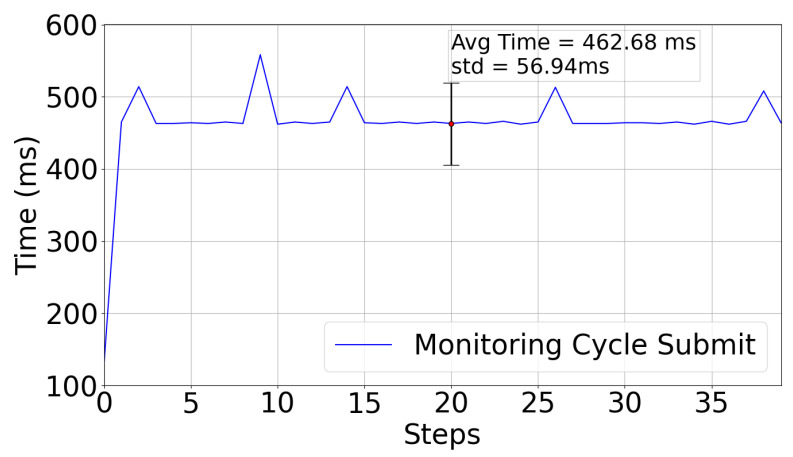
WaterMon Submission Time.

**Figure 17 sensors-22-07679-f017:**
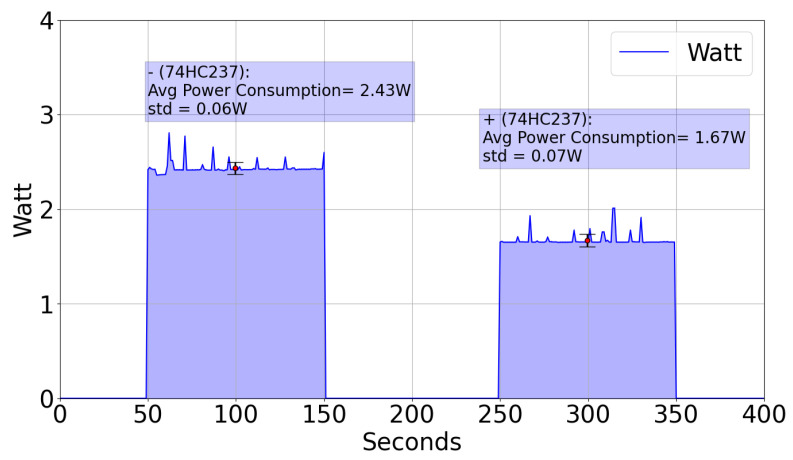
Power Consumption with and without IC 74HC237.

**Table 1 sensors-22-07679-t001:** Existing Remote IoT Sensor Models.

Model	pH	EC	DO	TDS	Temp	${\mathrm{NH}}_{4}^{+}$	ORP	${\mathrm{NO}}_{2}^{-}$	${\mathrm{NO}}_{3}^{-}$	Price (GBP)
TEKCOPLUS 6-in-1 Multi-Parameter [13]	X *			X			X *			80
GAIN EXPRESS 6-in-1 [14]	X			X	X		X			117
Bluelab Guardian Monitor [15]	X			X	X					265
Atlas Scientific Industrial Monitoring Kit [16]	X	X	X				X			1700
Hanna HI98194 [17]	X	X	X		X		X			2398
Nico 2000 ELIT Aqualiser [18]	X	X	X		X	X	X	X	X	3780

**Table 2 sensors-22-07679-t002:** Existing Wireless Sensor Models.

Model	pH	EC	DO	TDS	Temp	${\mathrm{NH}}_{4}^{+}$	ORP	${\mathrm{NO}}_{2}^{-}$	${\mathrm{NO}}_{3}^{-}$	Connect.	Price (GBP)
BIlinli PH-803W 2-in-1 PH ORP Controller [19]	X						X			WI-FI	100
Atlas Scientific Hydroponics Kit [20]	X	X			X					WI-FI	568
Libelium Smart Water [21]	X	X	X				X			Multiple	3800
Libelium Smart Water Extreme [21]	X	X	X		X		X			Multiple	5754
Libelium Smart Water Ions [21]						X		X	X	Multiple	4164

**Table 3 sensors-22-07679-t003:** Sensor Types and Monitoring Parameters.

	Type A	Type B	Type A + B	Minimal A + B
pH	X	X	X	X
EC	X	X	X	
DO	X		X	X
TDS				X
Temp	X		X	X
$NH_{4}^{+}$	X		X	*
$NH_{3}$		X	X	X
ORP	X		X	*
$NO_{2}$		X	X	*

**Table 4 sensors-22-07679-t004:** Technical specifications for Sensors.

	Meas. Range	Resolution	Accuracy	Response Time	Data Protoc.	Function.Temp
DO (AtlasSci.)	0.01–100 mg/L 0.1–400 % saturat.	0.01 mg/L	±0.05 mg/L	1 read per 420 ms	UART/I2C	−40 °C to 85 °C (Typ. 25 °C)
pH (AtlasSci.)	0.001–14.000	0.001	±0.002	1 read per 420 ms	UART/I2C	−40 °C to 85 °C (Typ. 25 °C)
EC (AtlasSci.)	0.07–500,000 uS/cm	0.07 uS/cm	±2%	1 read per 420 ms	UART/I2C	−40 °C to 85 °C (Typ. 25 °C)
ORP (AtlasSci.)	−1019.9–1019.9 mV	0.9 mV	±1 mV	1 read per 420 ms	UART/I2C	−40 °C to 85 °C (Typ. 25 °C)
Temp (AtlasSci.)	−126.000–1254 °C	0.001 °C	±(0.1 + 0.002 × °C)	1 read per 420 ms	UART/I2C	−40 °C to 85 °C (Typ. 25 °C)
$NO_{2}^{-}$ (Nico2000)	0.5 to 500 ppm ($10^{-5}$ Mol to 0.01 Mol)	0.5 ppm	Potential drift (in 1000 ppm): <3 mV/day (8 h)	<10 s	I2C	0 °C to 50 °C
$NO_{3}^{-}$ (Nico2000)	0.3 to 6200 ppm ($5\;\times \;10^{-6}$ Mol to 0.1 Mol)	0.3 ppm	Potential drift (in 1000 ppm): <3 mV/day (8 h)	<10 s	I2C	0 °C to 50 °C
$NH_{4}^{+}$ (Nico2000)	0.03 to 1800 ppm ($2\;\times \;10^{-6}$ Mol to 0.1 Mol)	0.03 ppm	Potential drift (in 1000 ppm): <3 mV/day (8 h)	<10 s	I2C	0 °C to 50 °C

**Table 5 sensors-22-07679-t005:** Parameters for LoRa configuration.

region = LoRa.EU868	selection of 868 MHz as carrier frequency, the default channel is established in 868.1 MHz
bandwidth = LoRa.BW_125KHZ	selection of a bandwidth of 125 kHz
sf = 7	sets the desired spreading factor; accepts values between 7 and 12
Device_class = LoRa.CLASS_C	Class C devices extend Class A by keeping the receive windows open, unless they are transmitting
coding_rate = LoRa.CODING_4_5	the default coding rate is 4/5
power_mode = LoRa.ALWAYS_ON	LoRa protocol is always listening
rx_iq = False	the use of IQ inversion allows mitigating interference between up- and downlink
LORA_PAYLOAD_SIZE = 240	payload size is configured at 240 bytes
LORA_TX_POWER = 7	this value establishes a modulation FSK, with 2 dBm

**Table 6 sensors-22-07679-t006:** Sensor’s Parameter dependencies for compensation.

Sensor	Parameter Dependency
EC	Temperature (RTC)
pH	Temperature (RTC)
DO	Temperature (RTC), Salinity (EC), Pressure (Fipy)
$NH_{4}^{+}$	Temperature (RTC), Salinity (EC),
$NO_{3}^{-}$	Temperature (RTC), pH
$NO_{2}^{-}$	Temperature (RTC), pH

## Data Availability

Not applicable.

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
