# Peer review of "Water IoT Monitoring System for Aquaponics Health and Fishery Applications"

_sensors, 2022, doi:10.3390/s22197679_

Round 1
Reviewer 1 Report
This manuscript describes the design and implementation details of IoT-integrated sensors for possible use in aquaponics health monitoring oriented to the fishery industry. The presented work fits well the scope of the sensors Journal. The manuscript is well-written and well-organized that addresses an important topic related to health monitoring of the fish farms industry. However, it needs extra effort to be improved and presented properly. In this regard, I have the following concerns that must be carefully considered by the authors.
1. The manuscript includes many abbreviated terms without any description. Therefore, any abbreviated term should be fully phrased before using it anywhere in the manuscript. There are many examples of that such as “WSN, LoRa, NB-LoRa, LoRaWAN, GSM, OTA, LTE, LTE-M, ……. etc.”.
2. Some figures (1 and 2) are not clear enough and need to be prepared in high resolution.
3. Considering Tables 1 and 2. The authors should mention the time of price collection as they are subject to changes with time.
4. The authors should mention the most important technical specifications of the used development board/microcontroller(s), and clarify how these specifications assisted or affected the results of their system implementation and performance.
5. The authors referred to many topics and terminologies, however, nothing can be found in the results section about these raised issues. Examples of that are LoRaWAN, SigFox, I2C protocol, opt-coupler, …. etc. These topics should be briefly described and clarified for the readers’ benefit.
6. The radio signal coverage was given in a unit of km^2. In practice, the range should be presented in km.
7. The implemented LoRaWAN technical specification should be mentioned in the relevant location of the manuscript, for example, what are the used frequency, channels, device classes, data rate, coding rate, payload, spreading factor, and link budget.
8. The authors did not discuss important issues regards their implemented IoT system such as latency, security, quality of service (QoS), packet-error rates, IoT platform protocols, and system reliability.
9. The authors didn’t discuss industry standards for the investigated sensors. Therefore, they should present a brief comparative study considering the most important measurement parameters of their implanted sensors and compare them to the industry standard. They should provide technical specifications for their equipment that define its range, accuracy, precision, resolution, and sensitivity. In addition, any precautions and special environmental conditions that influence the sensors' operation should be mentioned. Also, how the sensors' noise was identified, treated, and suppressed?
10. Regarding the implanted sensors’ probes/electrodes, the authors should mention the implementation and technical specification of these probes and how they were protected against corrosion and other severe environmental conditions in fish farms. Also, it is not clear whether the sensors are immersed in the water all the time or only during measurement times.
11. The authors discussed a lot the high cost of the industrial sensors and proposed a low-cost alternative system, however they didn’t finally mention the total cost of their final product.
Author Response
Reviewer 1
COMMENTS
This manuscript describes the design and implementation details of IoT-integrated sensors for possible use in aquaponics health monitoring oriented to the fishery industry. The presented work fits well the scope of the sensors Journal. The manuscript is well-written and well-organized that addresses an important topic related to health monitoring of the fish farms industry. However, it needs extra effort to be improved and presented properly. In this regard, I have the following concerns that must be carefully considered by the authors.
COMMENT 1
- The manuscript includes many abbreviated terms without any description. Therefore, any abbreviated term should be fully phrased before using it anywhere in the manuscript. There are many examples of that such as “WSN, LoRa, NB-LoRa, LoRaWAN, GSM, OTA, LTE, LTE-M, ……. etc.”.
ANSWER 1:
Thank you for this comment. We have added a list of acronyms at the end of the paper to make understandable the text.
COMMENT 2
- Some figures (1 and 2) are not clear enough and need to be prepared in high resolution.
ANSWER 2:
Thank you for this comment. Figures 1 and 2 have been improved to provide higher resolution.
Figure 1:
Figure 2:
COMMENT 3
- Considering Tables 1 and 2. The authors should mention the time of price collection as they are subject to changes with time.
ANSWER 3:
Thank you for this comment. The prices in these tables were collected from the time the links were accessed. At that time, we contacted the sellers to obtain the prices shown in the table. The final price of our proposal with all the sensors is around 1500£ per node, including communications. We have included such information in the new version of the manuscript.
COMMENT 4
- The authors should mention the most important technical specifications of the used development board/microcontroller(s), and clarify how these specifications assisted or affected the results of their system implementation and performance.
ANSWER 4:
Thank you for this comment. The Fipy node is based on ESP32 and its firmware have been modified directly from the Espressif-Xtensa. In the text, we have added the following references in section 6.1. Also we have included other specific information about the communication technologies. The selection of Fipy board was related to the versatility of this board in terms of the use of communication technologies and also the possibility to improve the implementation of the firmware due to the use of the ESP32 microcontroller. This could help in the implementation of energy consumption rules, combined with the IC.
“For the Gateway sensor, we will use a state-of-the-art ready-to-use sensor, specifically Pycom FiPy [29] with PySense 2.0 [30] extension boards. …”
- Pycom., 2022. Available at: https://docs.pycom.io/datasheets/development/fipy/ (Accessed: 22 September 2022).
- Pycom., 2022. Available at: https://docs.pycom.io/datasheets/expansionboards/pysense2/ (Accessed: 22 September 2022).
COMMENT 5
- The authors referred to many topics and terminologies, however, nothing can be found in the results section about these raised issues. Examples of that are LoRaWAN, SigFox, I2C protocol, opt-coupler, …. etc. These topics should be briefly described and clarified for the readers’ benefit.
ANSWER 5:
Thank you for this comment. Although this terminology is already well-known in the area of IoT and sensing, we have added some paragraphs to explain in which context we use them. They have been used mainly in the description of the design of the system. The main concepts used in the definition and design of our system have been briefly explained.
The term Sigfox does not appear in the text (just once in Figure 4, to show Fipy also works with this technology). However, we could add this text to the description: “Sigfox is also an LPWAN protocol which is Ultra Narrow Band(UNB)-based and it uses (D-)PSK or GPSK modulation. As well as LoRa, this protocol can achieve long ranges, but it is also ground penetrating.”
The term optocoupler does not appear explicitly in the text. However, we could add this text to the description: “The optocoupler, also named optical-coupler or opto-isolator, is an electronic component that transfers electrical signals between two isolated circuits by using light.”
In section 4, we have added these paragraphs:
“LoRaWAN is the specification for Low-Power Wide Area Networks (LPWAN) in Low Radiation (LoRa) communications. From the LoRa protocol, LoRaWAN is a point-to-multipoint network layer protocol, full duplex, and with the best LoRa features, i.e. low-power extending battery duration and quality of service (with 3 modes), and uses FSK modulation. LoRa-based technology is able to achieve long ranges.
LTE-M (or LTE-MTC [Machine Type Communication]) including eMTC (enhanced Machine Type Communication), is a low-power wide area network (LPWAN) radio technology standard developed by 3GPP to enable a wide range of cellular devices and services (specifically, for machine-to-machine and Internet of Things applications). The specification for eMTC (LTE Cat M1) was frozen in 3GPP Release 13 (LTE Advanced Pro), in June 2016.
NB-IoT is also an LPWAN protocol, and it is a radio technology standard developed by 3GPP to enable high-end services for mobile devices. Previously, the 3GPP Release 13 (LTE Advanced Pro) specification was used until June 2016. NB-IoT uses an extension of the standard LTE network but limits bandwidth to a single 200kHz band. Uses OFDM modulation for communications with few connections between devices and uses SC-FDMA for communications with more connections between devices.
The advantage of LTE-M over NB-IoT is its comparatively higher data rate, mobility, and voice over the network, but it requires more bandwidth and is more costly. Compared to LTE Release 12 Cat-0 modem, an LTE-M model is claimed to be 80% less expensive, supports up to 18 dB better coverage, and has a battery life-time that can last up to several years.”
In subsection 5.1, we have added the following paragraph:
“I2C or Inter-Integrated-Circuit protocol is a synchronous, multi-controller/multi-target, packet-switched, single-ended, serial communication bus. It is widely used for attaching lower-speed peripheral ICs to processors and microcontrollers in short-distance, intra-board communication.”
COMMENT 6
- The radio signal coverage was given in a unit of km^2. In practice, the range should be presented in km.
ANSWER 6:
Thank you for this comment. The coverage is referred to as the coverage area of the aquaponic farm, for this reason, we use km^2, instead of km.
COMMENT 7
- The implemented LoRaWAN technical specification should be mentioned in the relevant location of the manuscript, for example, what are the used frequency, channels, device classes, data rate, coding rate, payload, spreading factor, and link budget.
ANSWER 7:
Thank you for this comment. We have used the specification of Fipy to implement the network with LoRa, and the most common frequency in Europe is 868MHz. Also, the specification for LoRa in Fipy, with class C (although the specification also allows class A)
In the code, LoRa protocol has been specified as:
Lora = network.LoRa(mode=LoRa.LORA, region=LoRa.EU868, power_mode=LoRa.ALWAYS_ON, bandwidth=LoRa.BW_125KHZ, sf=7, device_class=LoRa.CLASS_C, rx_iq=False)
The properties are:
region=LoRa.EU868 (specifies the European region, using 868MHz as carrier frequency, the default channel is established in 868.1 MHz)
bandwidth = LoRa.BW_125KHZ
sf=7 (sets the desired spreading factor. Accepts values between 7 and 12, but in our case, we have used the minimum spreading factor)
Device_class = LoRa.CLASS_C (Class C devices extend Class A by keeping the receive windows open unless they are transmitting. This allows for low-latency communication but is many times more energy-consuming than Class A devices.)
coding_rate=LoRa.CODING_4_5 (the default coding rate is 4/5)
power_mode=LoRa.ALWAYS_ON
rx_iq=False (without IQ inversion. The use of this technique allows to mitigate interference between uplink and downlink, but also is)
Also:
LORA_PAYLOAD_SIZE=240
LORA_TX_POWER=7 (this value establishes a modulation FSK, with 2 dBm)
This specification and definition can also be obtained from Pycom’s website (https://docs.pycom.io/firmwareapi/pycom/network/lora/#app)
We have added the following paragraph and table:
For the implementation of the LoRa protocol, we have selected the parameters shown in Table 4
COMMENT 8
- The authors did not discuss important issues regarding their implemented IoT system such as latency, security, quality of service (QoS), packet-error rates, IoT platform protocols, and system reliability.
ANSWER 8:
Thank you for this comment. It is true that we didn’t make latency tests, neither packet-error tests, nor system reliability tests, as this is a first study of the system. Our purpose here is to describe the implementation and the functioning test.
IoT platform protocols have been adequately described, as we understand that communication technologies (LoRa/LoRaWAN, WiFi, NB-IoT, LTE-M) and MQTT are protocols involved in the IoT platform.
Regarding security issues, we can add some security issues related to the MQTT packets. The publication and subscription policies are configured to level 2 for both subscribers and publishers in order to ensure delivery and no duplication. Also in relation to these security issues, MQTT brokers generally provide Transport Layer Security (TLS) encryption for secure communication between clients and brokers.
We have added the following paragraph in section 4:
“The data link from the node to the farmhouse is made using Message Queuing Telemetry Transport (MQTT) protocol. This protocol can be configured with 3 levels (i.e. 0, 1 and 2), but for our purpose, it has been configured with level 2 for both subscribers and publishers to ensure information delivery and no data duplication. The security issues were tackled by configuring the Transport Layer Security (TLS) encryption to ensure secure communication between clients and brokers.”
Regarding QoS, our solution does not support any kind of QoS at MAC layer. It provides the basic QoS available in IP Layer and also the QoS capabilities provided by the MQTT protocol. But these are obvious aspects already existing in literature.
Regarding error-rate, the system is performing a real-time continuous monitoring and it has implemented at application layer the re-delivery of packages, thus error-rate is not a concern, at least within the coverage area tested and validated.
COMMENT 9
- The authors didn’t discuss industry standards for the investigated sensors. Therefore, they should present a brief comparative study considering the most important measurement parameters of their implanted sensors and compare them to the industry standard. They should provide technical specifications for their equipment that define its range, accuracy, precision, resolution, and sensitivity. In addition, any precautions and special environmental conditions that influence the sensors' operation should be mentioned. Also, how the sensors' noise was identified, treated, and suppressed?
ANSWER 9:
Thank you for this comment. The technical specifications for the sensors included in the node are shown in the following table:
|
|
Meas. Range |
Resolution |
Accuracy |
Response time |
Data protocol |
Function.Temp |
|
DO (AtlasSci.) |
0.01-100 mg/L 0.1 - 400 % saturation |
0.01 mg/L |
± 0.05 mg/L |
1 read every 420ms |
UART/I2C |
-40 to 85ºC (Typ.25ºC) |
|
pH (AtlasSci.) |
0.001-14.000 |
0.001 |
± 0.002 |
1 read every 420ms |
UART/I2C |
-40 to 85ºC (Typ.25ºC) |
|
EC (AtlasSci.) |
0.07-500,000 uS/cm |
0.07 uS/cm |
± 2% |
1 read every 420ms |
UART/I2C |
-40 to 85ºC (Typ.25ºC) |
|
ORP (AtlasSci.) |
-1019.9 - 1019.9 mV |
0.9 mV |
± 1 mV |
1 read every 420ms |
UART/I2C |
-40 to 85ºC (Typ.25ºC) |
|
Temp (AtlasSci.) |
-126.000 - 1254 ºC |
0.001 ºC |
± (0.1 + 0.002 x ºC) |
1 read every 420ms |
UART/I2C |
-40 to 85ºC (Typ.25ºC) |
|
NO2- (Nico2000) |
0.5 to 500ppm (1x10-5 to 0.01 Molar) |
0.5 ppm |
Potential drift (in 1000 ppm) : < 3 mV/day (8 hours) |
< 10s |
I2C |
0 to 50ºC |
|
NO3- (Nico2000) |
0.3 to 6,200 ppm (5x10-6 to 0.1 Molar) |
0.3 ppm |
Potential drift (in 1000 ppm) : < 3 mV/day (8 hours) |
< 10s |
I2C |
0 to 50ºC |
|
NH4+ (Nico2000) |
0.03 to 1,800 ppm (2x10-6 to 0.1 Molar) |
0.03 ppm |
Potential drift (in 1000 ppm) : < 3 mV/day (8 hours) |
< 10s |
I2C |
0 to 50ºC |
Noise typically affects analog devices. We have designed the motherboard keeping a complete layer of the board only for analog devices to supress at maximum the noise. In our case, the analog-to-digital converter (ADC) is also provided with the probe, so the noise is processed in this ADC. All the sensors considered are already digitalised for data transmission. In the case of NO2-, NO3- and NH4+, the electronic has been protected with an optocoupler, in order to avoid interference between signals within these probes.
COMMENT 10
- Regarding the implanted sensors’ probes/electrodes, the authors should mention the implementation and technical specification of these probes and how they were protected against corrosion and other severe environmental conditions in fish farms. Also, it is not clear whether the sensors are immersed in the water all the time or only during measurement times.
ANSWER 10:
Thank you for this comment. Most of the sensors are made with a teflon membrane, but they have also been provided with a capsule into which water is introduced. Obviously, over time these sensor probes can have degradation, but not as much as with direct exposure of the sensors to the water in the ponds. Also, the electrodes can be cleaned with a filling solution (electrolyte) with a fine polishing cloth and rinsed with distilled water and alcohol. They require constant maintenance every month and they replacement every year as part of the daily erosion in the operational environment.
COMMENT 11
- The authors discussed a lot the high cost of the industrial sensors and proposed a low-cost alternative system, however they didn’t finally mention the total cost of their final product.
ANSWER 11:
Thank you for this comment, the final cost for the complete node is 1500 GBP (considering all the sensor probes, the electronics and the communications).

Reviewer 2 Report
I have the following comments:
1. The organization of paper need to be improved
2. The technical merits need to be improved
3. Proposed IoT System Architecture is not clear and does not follow IoT Methodology
4. Results for IoT implementation is not available
5. Mathematical models for channel characteristics need to be captured
Author Response
Reviewer 2
COMMENTS
I have the following comments:
COMMENT 1
- The organization of the paper need to be improved
ANSWER 1:
Thank you for this comment. The organization of the paper is based on the description of the development done. In order to improve the readability and comprehension of the paper, we have added some technical information about the devices and subsystems used.
COMMENT TO THE EDITOR:
Please, have in mind the Reviewer 1 said: “The manuscript is well-written and well-organized”. Thus, we have done our best to enhance the read-ability and organization of the reviewed version of the manuscript.
COMMENT 2
- The technical merits need to be improved
ANSWER 2:
Thank you for this comment. We have improved the technical information in the paper, by adding information about the different sensors involved. The paper is oriented to describe the development of an IoT system for Aquaponics control and management. The development itself has some technical merits, but also the improvement of the energetic consumption in this system is a considerable technical improvement.
COMMENT 3
- Proposed IoT System Architecture is not clear and does not follow IoT Methodology
ANSWER 3:
Thank you for this comment. It has allowed as to realize that we have not indicated the IoT Methodology we are using, to be concrete, the one standardized by ITU-T under specification ITU-T Y.2060. We have included the following text in the manuscript:
In our proposed architecture, we have designed and prototyped both the device and gateway capabilities indicated in the IoT Methodology standardized by ITU-T under specification ITU-T Y.2060 [ 28 ] . However, at this stage, we have not specified the definition of the functional components defined in ITU-T Y.4115-2017 [29 ] as our IoT system is a industrially relevant lab-based development, so the IoT Device Capability Exposure (IoT DCE) is summarised in a Database and a Dashboard (in the Farmhouse subsystem), but could be replaced by a complete IoT ecosystem, such as Thingsboard, FIWARE or any other platform.
- ITU. Next Generation Networks – Frameworks and functional architecture models. https://www.itu.int/ITU-T/ recommendations/rec.aspx?rec=y.2060, 2012. (Accessed 29-Sep-2022).
- ITU. Reference architecture for IoT device capability exposure. https://www.itu.int/rec/T-REC-Y.4115/en, 2012. (Accessed 29-Sep-2022).REC-Y.4115/en. [Accessed 29-Sep-2022].
COMMENT 4
- Results for IoT implementation is not available
ANSWER 4:
The functionality of the IoT implementation has been designed, prototyped, manufactures, verified, tested and validated. Several implementation results have been provided in the associated section of the manuscript. We wonder why the reviewer have provide such comment and what results are the ones referred by the reviewer. If you are referring to the fact that the source code is not available. You are absolutely right as we are currently in the process of creating an Spin-off to initiate their commercialization.
COMMENT 5
- Mathematical models for channel characteristics need to be captured
ANSWER 5:
This comment does not make sense in a journal entitled “Sensors”. It is a comment more related to a journal really focused on “Telecommunications” and “Antenna Propagations”. Having said that, please have in mind that we are using standardized radio interfaces and each of them have of course their associated channel characteristics in the manufacturer of such communications. Thus, we have provided the concrete details of the modems being used, allowing the reader to get any channel characteristics from the manufacturer if it is of interest, including any mathematical model or similar documentation.
In order to do our best to address this comment. We have added the information related to the low-level configuration of LoRa interface to allow any reader to be able to reduce our research with publication the concrete configuration parameters of the radio interface. Please, note that this is not a theoretical paper and but a real laboratory-based deployment, We have added this table:
For the implementation of the LoRa protocol, we have selected the parameters shown in Table 4

Round 2
Reviewer 1 Report
Accepted in present form
Reviewer 2 Report
The paper may be accepted